# Enhanced Multiple-Object Tracking Using Delay Processing and Binary-Channel Verification

**Muyu Li [1,2,3,4,*,†]**, **Xin He [1]**, **Zhonghui Wei [1]**, **Jun Wang [1]**, **Zhiya Mu [1]** and **Arjan Kuijper [3,4,*]**

[1] Changchun Institute of Optics, Fine Mechanics and Physics, Chinese Academy of Sciences, Changchun 130033, China; hexin6627@sohu.com (X.H.); wzhlvp@sohu.com (Z.W.); zif1005@163.com (J.W.); muziya9@163.com (Z.M.)
[2] University of Chinese Academy of Sciences, Beijing 100049, China
[3] Fraunhofer IGD, Darmstadt, 64285, Germany
[4] TU Darmstadt, Darmstadt, 64285, Germany; arjan.kuijper@tu-darmstadt.de
* Correspondence: muyu.li@gris.tu-darmstadt.de or limuyu14@mails.ucas.ac.cn (M.L.); arjan.kuijper@igd.fhg.de (A.K.)
† Current address: Fraunhofer IGD, Darmstadt, 64285, Germany.

**Abstract:** Tracking objects over time, i.e., identity (ID) consistency, is important when dealing with multiple object tracking (MOT). Especially in complex scenes with occlusion and interaction of objects this is challenging. Significant improvements in single object tracking (SOT) methods have inspired the introduction of SOT to MOT to improve the robustness, that is, maintaining object identities as long as possible, as well as helping alleviate the limitations from imperfect detections. SOT methods are constantly generalized to capture appearance changes of the object, and designed to efficiently distinguish the object from the background. Hence, simply extending SOT to a MOT scenario, which consists of a complex scene with spatially mixed, occluded, and similar objects, will encounter problems in computational efficiency and drifted results. To address this issue, we propose a binary-channel verification model that deeply excavates the potential of SOT in refining the representation while maintaining the identities of the object. In particular, we construct an integrated model that jointly processes the previous information of existing objects and new incoming detections, by using a unified correlation filter through the whole process to maintain consistency. A delay processing strategy consisting of the three parts—attaching, re-initialization, and re-claiming—is proposed to tackle drifted results caused by occlusion. Avoiding the fuzzy appearance features of complex scenes in MOT, this strategy can improve the ability to distinguish specific objects from each other without contaminating the fragile training space of a single object tracker, which is the main cause of the drift results. We demonstrate the effectiveness of our proposed approach on the MOT17 challenge benchmarks. Our approach shows better overall ID consistency performance in comparison with previous works.

**Keywords:** multiple object tracking; identity consistency; single object tracking

## 1. Introduction

Multiple object tracking (MOT) in video (a critical problem for many applications including robotics, video surveillance, and autonomous driving) remains one of the big challenges of computer vision. The goal is to locate all the objects we are interested in, in a series of frames, and form a reasonable trajectory for each one of them. Since recent progress has been made on object detecting, tracking-by-detection (shown in Figure 1) has emerged as one of the most popular paradigms to solve the MOT problem, as it breaks MOT into two parts: (i) detection, to separate interesting objects from

the background in each frame, and (ii) data association, to align corresponding detections through time series to form reasonable trajectories. However, missing and spurious detections, as well as complex scenes under a MOT scenario like occlusions and object interactions in crowded environments (some of the examples are shown in Figure 1), may set a barrier for the performance of MOT methods due to the high dependency of detection. Also, the separation of detection from tracking itself may keep object detection inaccessible to the temporal information of a certain object.

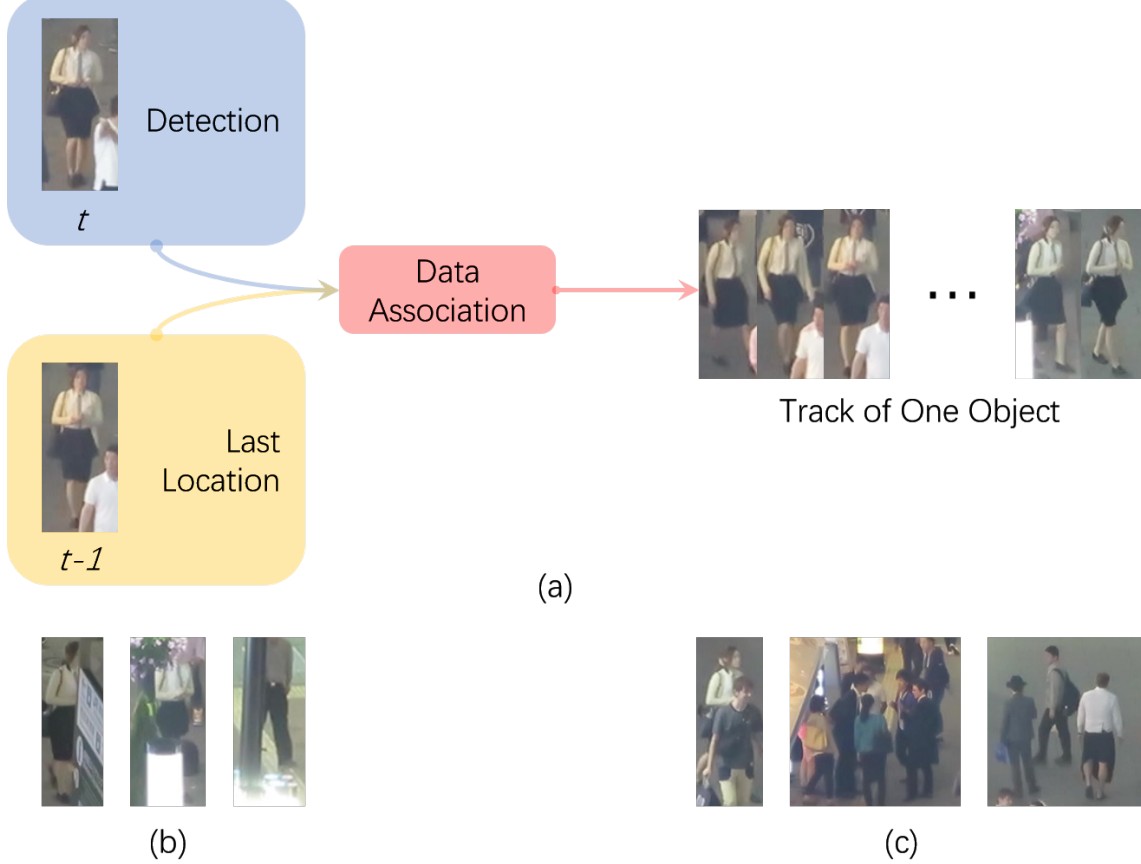

**Figure 1.** (**a**) One example of the overall tracking-by-detection paradigm in multiple object tracking (MOT). At time *t*, the MOT method matches the last location of a certain object and aligns the track with the upcoming detection provided by an external detector. Eventually, the goal is to form a reasonable track of that object by repeating this process throughout the video. (**b**) Some examples of occlusion under a MOT scenario. (**c**) Some examples of object interaction.

A common concern in MOT approaches is identity (ID) consistency—maintaining the objects' identities as long as possible. This is difficult under occlusion and in crowded scenes, especially when the appearances of the individual objects are not distinctive enough. Some of the offline methods rely on tracklet (short trajectory segments) merging, rather than single detections, to keep track of the objects. Online methods, on the other hand, use a re-identification method to retrieve certain objects which are lost from tracking during occlusion or interaction. However, their performance is still not as promising as desired. A single object tracker is capable of distinguishing objects from the background, where the goal is to maintain tracking a single object as long as possible. Hence, it is intuitive to implement a single object tracking (SOT) method in a MOT scenario to keep the ID consistency.

The MOT problem can be easily rephrased as multiple single object tracking problem, where all of the objects' states are estimated by a tracker formed of multiple single object trackers. However, scenes under a MOT scenario are quite different from those in a SOT one, usually they are more complex. Thus, directly pushing SOT methods into a MOT scenario still faces various challenges.

The aim of single object tracking (SOT) is to locate an object in continuous video frames given an initial annotation in the first frame. In contrast with MOT, there is only one object of interest and it is always in the considered image scene. A single object tracker is constantly generalized to capture appearance changes of the object, and it is designed to efficiently distinguish the object from the background by training a strong discriminative appearance model to find the location of the object within a searching area in the next frame. However, in the MOT context, multiple objects with similar appearances and geometries in the searching area may confuse the single object tracker. Furthermore, the update strategy of the single object tracker may lead to an ambiguity problem. These kinds of noisy samples may contaminate the online training samples for the tracker, can lead to gradual drift, and eventually will fail to track the object. Moreover, since the steady tracking of the single object tracker heavily relies on the quality of initialized bounding-box in the first frame in order to correctly separate the object from the background and possibly retrieve it after occlusion, the object candidates provided by a real detector under the current MOT framework are usually imperfect. With considerable noise in location and scale, this may become another major challenge.

In this paper, we propose an integrated model with a binary-channel verification and delay processing model to tackle the aforementioned problems; see Figure 2 for an outline. When a new frame comes in, SOT trackers produce two scores on both the predicting location given by the motion model (the same as processing under the SOT scenario) and the region of interest provided by the detection. Combined with the previous tracking score and the trajectory, the association model produces a refined bounding box. When occlusion happens, the association model proceeds as usual, then marks the frame ID where the occlusion happens and keeps the results for later processing. A delay processing model composed with three processing parts is proposed. First, for those tracks encountering the deformation problem (as in those with a low tracker score and low detection score), re-initiation discards the non-deformed features trained before and re-initiates with new features after determination by spatial information as the same object. Second, attaching for newly initialized tracklets during the occlusion period, re-correlates them with tracks lost in the same period and re-attaches them if the correlation score is above a certain threshold. Third, re-claiming, for unattached newly initialized tracklets after the attach phase , assigns them to existing tracks in shadow tracking mode, which means the object is tracked without any verification of detection through recent frames, if the re-correlation score and overlap rate (as in the Intersection over Union (IOU) score between the newly initialized tracklets and existing tracks for several frames) between them are above specified thresholds.

This paper thus presents three main contributions:

- We propose a binary-channel verification model to deeply excavate the potential of applying SOT under a MOT scenario, specifically in refining the representation while maintaining the identities of the objects during the tracking process.
- We introduce a delay processing model to improve the tracker capability of solving the drift problem under occlusion and object interaction.
- We explore the proposed method on the MOT17 benchmark and perform an ablation study of each block, showing the comparable performance on result-refinement and ID consistency, and proving each block of our proposed method is indispensable.

This paper is organised as follows. Section 2 briefly reviews the state-of-the-art for computer vision-based object tracking. The details of our proposed method are described in Section 3. The experimental study and benchmark evaluation are then reported in Section 4. Finally, Section 5 concludes this paper and describes opportunities for future research.

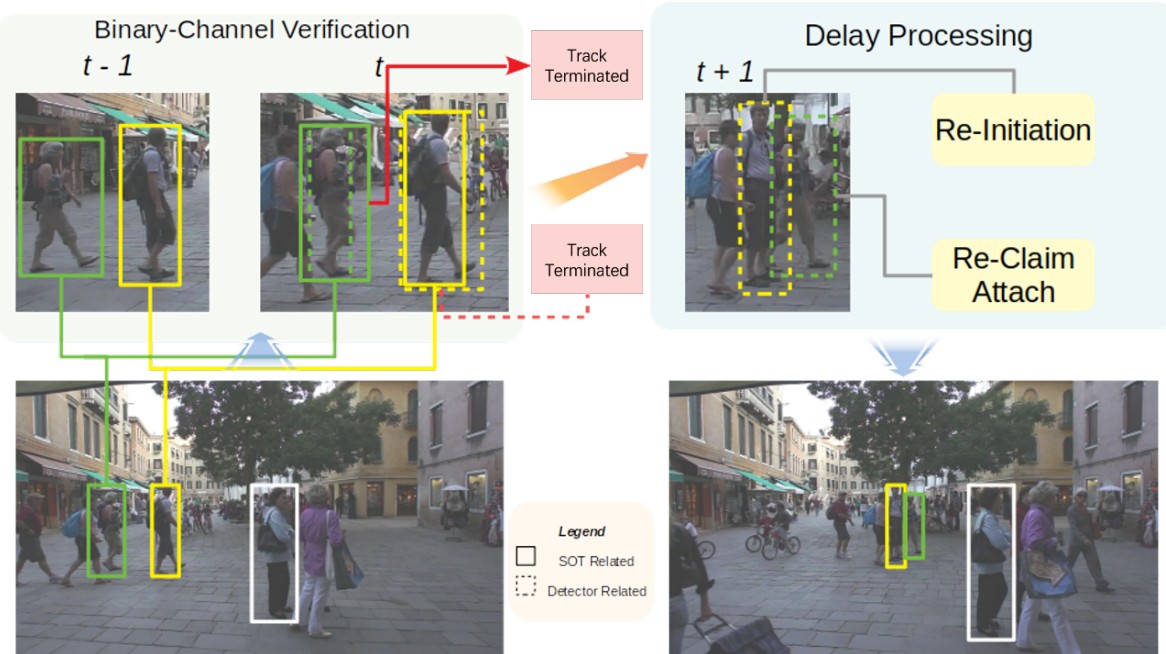

**Figure 2.** An example of our method. At time *t*, the estimated bounding boxes provided by every single object tracker are sent to the binary-channel verification model along with the bounding boxes from the external detector, where the determination of the output is based on the evaluation score of the single object tracker alone. Then, a delay processing model focusing on potential complex circumstance like *t* + 1 above is proposed to handle occlusion, deformation, etc. Single object tracking (SOT).

## 2. Related Work

### 2.1. Multiple Object Tracking

Although a great deal of work has been done on multiple object tracking (MOT), it still remains a challenging task, especially in complex environments where occlusions and imperfect detections are common. Recent works on MOT primarily focus on the tracking-by-detection paradigm. Most of them can be roughly categorized into offline tracking and online tracking.

In offline tracking, MOT is formed as an offline global optimization problem which uses frame observation through previous and future states to estimate the current status of objects [11–15]. A data association method is widely used, such as the Hungarian algorithm [16,17], network flow [18,19], or multiple hypotheses tracking [20]. However, offline methods are not suitable for causal applications like autonomous driving, since future information is necessary.

Things are quite different in online tracking, where observations from before the current frame are provided to the online estimation of object states. Trajectories are generated based on information only up to the current frame, which adopt probabilistic inference [21] or deterministic optimization. However, online methods heavily rely on the performance of the underlying detector, since they are more sensitive to noisy detections. This is a resource waste of the tracker itself, since the detector is unable to access the temporal information from the previous trajectories. Our work focuses on extending online single object tracking (SOT) methods to MOT scenarios. SOT helps to alleviate the limitations from imperfect detections, especially for missing detections. It is complementary to the data association methods, since the tracking results of single object trackers at the current frame can be considered as association candidates for the data association.

### 2.2. Single Object Tracking Based on Regularized Correlation Filters

Correlation filters (CF) have been actively adopted in single object tracking to improve robustness and efficiency. Initially, since CF needs training, the work of online tracking seems inappropriate. In the later years, with the development of the minimum output of sum of squared error (MOSSE) filter [28], which introduces efficient adaptive training, changed the situation. To meet the real-time requirement, a high-speed tracking method using kernelized correlation filters (KCF) [29] has been proposed, using a Gaussian kernel function to reduce computational complexity. However, discriminative correlation filter (DCF) based trackers are limited in their detection range because of the requirement of equal filter and patch size, which leads to a constant failure under occlusion and object deformation. Thus, a larger search region is necessary, which needs incorporation of a measure of regularization due to the degradation of the discriminative power under more complex background. Danelljan et al. [44] presented spatially regularized DCF (SRDCF) by introducing a spatial regularization in DCF learning. To improve the performance of SRDCF, Danelljan et al. [5] replaced the Histogram of Oriented Gradient (HOG) features to deep features in a Convolutional Neural Network (CNN) . Then, in [30], a multi-resolution feature maps learning, named continuous convolutional operators for tracking (C-COT), was proposed to optimize the estimation of the object position. Finally, the efficient convolution operators (ECO) tracking scheme [3], as an improved version of C-COT, was proposed. ECO picks the most efficient set of filters and discards unnecessary ones, and uses a Gaussian mixture model (GMM) to represent diverse object appearances. These methods led to increasing performance of the tracker and significantly reduced processing time.

### 2.3. A Single Object Tracker in MOT

Some of the previous MOT methods have attempted to adopt SOT methods into the MOT problem. However, SOT methods are often used to tackle sub-problems. For instance, in [7], SOT is only used to generate initial tracklets. Yu et al. [8] partitioned the state space of the object into four subspaces and only utilized single object trackers to track objects in a tracked state. Chu et al. [1] implement the single object tracker as a state estimation procedure and further train a CNN based classifier to handle re-tracking and occlusion problems.

Few works integrated SOT methods throughout the whole tracking process. Breitenstein et al. [9] trained an object-specific classifier to compute the similarity for data association in a particle filtering framework. Yan et al. [4] kept both the tracking results of the single object trackers and the object detections as association candidates and selected the optimal candidate using an ensemble framework. However, tracking drift caused by occlusion still remains unsolved.

Chu et al. [10] used a dynamic CNN-based framework with a learned spatial-temporal attention map to handle occlusion, where the CNN trained on ImageNet is used for pedestrian feature extraction. Although it showed a good tracking performance, memory and time consumption may explode, since it assigns a network for each object and conducts online learning. Our approach differs from these methods by using a SOT method with a two-step verification and delay processing strategies to refine the tracker results and tackle the drift problem, in order to deeply excavate the potential of applying SOT under MOT scenarios.

## 3. Methodology

### 3.1. Overview

An overview of our proposed approach is shown in Figure 2. Following the tracking-by-detection paradigm, the online MOT can be formulated as an optimization problem. Trajectories in the set $\mathbb{T} = \{track_1, track_2, \cdots, track_k\}$ are forked by the bounding boxes $B_k = \{x, y, w, h\}$ of every object in

the past frames. At frame $t$, the set of the objects' locations $\mathbb{B} = \{b_{k_1}, b_{k_2}, \cdots, b_{k_n}\}$ of the previous frame $t$-$1$ are associate with the $\mathbb{D}^t$ detection candidates to maximize a final score:

$$B_k = \operatorname{argmax} f_k \left( t, B_k^{t-1}, D_j^t, \mathcal{A}^t, \mathcal{W}_{v,d_{k,j}}^t \right). \tag{1}$$

Here, $\mathcal{A}^t$ denotes the final association results between the *k-th* object in $B$ and the *j-th* detection in $D$ at frame $t$. $\neg_{k_j}^t = 1$ indicates that the association of track $k$ and detection $j$ is valid, while $\neg_{k_j}^t = 0$ otherwise. Furthermore, $\neg_{k_j}^t = 0.5$ means that frame $t$ is in a complex environment and the association result needs to be post-process by the delay processing model. $\mathcal{W}$ is a set of parameters to model each object's state, including previous information, deformation, and interaction status, which is learned using the appearance and location information through previous frames. The function $f(*)$ calculates the overall results of the tracking procedure for all the existing objects at frame $t$, defined as follows:

$$f(t, B_k) = \sum_{k_j} \mathcal{A}_{s_T, s_O, k_j}^t \cdot SOT \left( t, b_k^{t-1}, d_j^t, \sqsupseteq_{v,d,k_j}^t \right). \tag{2}$$

Therefore, solving the online MOT problem is done by solving Equation (1) from frame to frame.

### 3.2. Binary-Channel Verification Model

A binary-channel verification model is proposed to improve the quality of the state estimation as function of each object. For each object in frame $t$, a function $SOT_k$ is learned to separate the object itself from the background information by assigning a high score to it, while returning low scores for the background. The parameter $\neg_{k,j}^t$ is set to choose the best correlated spatial location $d_j^t$ from $D^t$ on object $k$ exclusively to one object only referring to scores from all objects. Regarding different states of incoming tracks, ongoing tracks and occluded tracks, this model can operate on those two tracking states in a similar way with some slight changes.

The function $SOT_k$, as in the state estimation, is the first step to process the incoming frame. An ordinary single object tracker is initialized by the object groundtruth bounding box in the first frame where a certain object existed, and is slowly and constantly updated. However, in MOT, the tracker is initially learned from detections provided by an external detector which contains considerable noise with respect to the location. Moreover, if the object moves closer or away from the viewing point, the scale of the object will also change rapidly. Hence, implementing a single object tracker under a MOT scenario without any supervision will easily lead to identity switches or catastrophic mis-track problems. Benefiting from the tracking-by-detection paradigm, detection can play the role of the supervisor for the tracker. During the state estimation process, we rewrite the function as:

$$SOT_k = \left[ SOT_T = \left( t, b_k^{t-1}, d_j^t, {}_{v,d,k_j}^t \right), SOT_D = \left( t, b_j^t, d_j^t, {}_{v,d,k_j}^t \right) \right]. \tag{3}$$

For subsequent tracks, $SOT_k$ estimates the score $s_{T_k}$ on the previous location containing the object $k$ (we assume the object has moved only slightly between frames, which is verified by high frame rates), while $SOT_D$ calculates the score $s_{D_j}$ on the location of the detection $j$ within the search area using the same trained single object tracker. $s_{T_k}$ shows the confidence of the estimated bounding box provided by the tracker as well as the probability of the object $k$'s existence in the current frame, while $s_{D_j}$ gives

the similarity between the detection $j$ and the object $k$. The output bounding box can be assigned as follows:

$$
b_k = \begin{cases}
b_{det} & if\ s_{D_j} > s_{T_k}\ and\ s_{T_k}, s_{D_j} > s_{BCVthreshold} \\
b_{tracker} & if\ s_{D_j} < s_{T_k}\ and\ s_{T_k}, s_{D_j} > s_{BCVthreshold} \\
'loss' & if\ s_{T_k} < s_{BCVthreshold} \\
'misaligned' & if\ s_{D_j} < s_{BCVthreshold}\ and\ s_{T_k} > s_{BCVthreshold}
\end{cases}
$$ (4)

$$
where \quad s_{T_k} = SOT_T\left(t, b_k^{t-1}, d_{j,v,d,k_j}^{t,t}\right), s_{D_j} = SOT_D\left(t, b_j^t, d_{j,v,d,k_j}^{t,t}\right).
$$

Here, $s_{BCVthreshold}$ is based on previous tracking information, $b_k^{t-1}$ represents the bounding box of the same object in the previous frame and $b_j^t$ for the bounding box of the $j$-th detection in the current frame. Also, results with an untrack mark will be sent to the memory waiting for future re-claim, in order to save computational resources. For occluded tracks, $b_k^{t-1}$, since it did not exist, the estimation of the object should take place at a search area provided by the motion model instead of the previous location, for which we follow the constant velocity assumption (CVA) [21,22] with a camera motion compensation by calculating the motion information of highly trusted objects' motion trends.

We take advantage of the idea of the non zero-sum game [27], where scores calculated on the estimation of the tracker and associated detection push each other from frame to frame with the concept that the result can be optimized as long as one of them shows a promising result, in order to better refine the output bounding box and increase the quality of online training samples added to the tracker at each frame. In this way, the single object tracker and the detection can be complementary to each other and improve the performances of each other along the tracking process.

*3.3. Delay Processing Model*

SOT methods are designed to distinguish the object from the background and allow a certain degree of deformation to handle appearance variation of the object. This mechanism will easily lead to insensitivity of the tracker when distractions are similar to the object within the search area under the MOT scenario. Hence, directly pushing a SOT method to MOT may easily lead to drifting problems. To tackle the drifting situation caused by occlusion or instance interaction, we propose a delay processing model. The consequences of occlusion can be seen as three main circumstances:

1. Object deformation. This appears when $s_{T_k}$ and $s_{D_j}$ are at the same level but below the association threshold $s_{DPthreshold}$. This kind of circumstance is usually determined as lost by the single object tracker, which leads to the ID switch problem.
2. Object misalignment. Here the object track is assigned to a false detection with similar appearance and motion features, usually expressed as an abrupt change of location, receiving a reasonable estimation score $s_{T_k}$ and similarity score $s_{D_j}$.
3. Doppelganger initialization. This occurs if the object experiences partial occlusion or slight variation of appearance, where the tracker finds a new location at the current frame without verification of any detection and keeps tracking for upcoming frames. Hence, a doppelganger track will be initialized using the detection supposed to be assigned to the object, causing an overlap problem.

The three circumstances set $\mathbb{C}$ is thus formed as follows:

$$
\mathbb{C} = \begin{cases}
'deformed' & if\ s_{T_k} \in s_{D_j} \cdot (1 \pm \sigma) < s_{DPthreshold} \\
'misaligned' & if\ s_{T_k} \in s_{D_j} \cdot (1 \pm \sigma) > s_{DPthreshold} \\
'doppelganger' & if\ s_{T_k} > s_{DPthreshold}, s_{D_j} > s_{DPthreshold}
\end{cases}
$$ (5)

Here, $\sigma$ is set to 0.2. To solve the aforementioned problems, we propose a delay processing model consisting of three parts which specifically target those circumstances. Such model is not only able

to avoid the noisy appearance features that might contaminate the training space of the single object tracker, but to change the update strategy of the tracker to ensure the tracking performance.

### 3.3.1. Re-Initiation

The first part is called Re-Initiation, as shown in Figure 3. When the object encounters the deformation problem, *e.g.* a pedestrian turning around, appearance models tend to fail – including the single object tracker. It is non-trivial to rely on motion estimation model to determine the location with highest probability after occlusion for certain objects. We follow the constant velocity assumption and assume that the object's velocity can not change abruptly through neighboring frames. If the object is retrieved by the motion model and validated by the single object tracker using an independent retrieving score $s_{retrieve}$ after several occlusion frames, the tracker discards the outdated non-deformate appearance features and re-initiates with new samples.

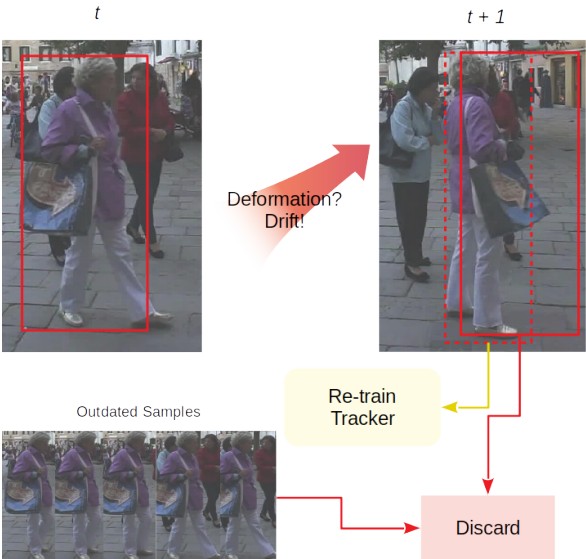

**Figure 3.** An illustration of the re-initiation phase in the proposed delay processing model. When deformation happens in an upcoming timestamp $t + 1$, the single object tracker may suffer the drift problem as shown in the solid box. The proposed method will re-train the single object tracker using a new detection (denoted by the dotted box) and discard outdated appearance features.

### 3.3.2. Attaching

The second part is named attaching (see Figure 4). When an object is misaligned with a false detection, the appearance model of the single object tracker is still powerless, since distraction with a similar appearance and geometry can easily give the single object tracker an illusion which may lead to drift results. However, with the high frame rate of the sequence, we assume that the object has moved only slightly between frames. Following this assumption, misaligned distraction can be notified if the offset of the object between adjacent frames is beyond a noticeable score based on the recent motion trend. If this happens, the wrong association is discarded and the object is marked as occluded for now, waiting for later processing. When the occlusion stops, several new tracks may be initialized due to the above situation, the tracker will perform a second-time correlation between the new tracks and the occluded tracks labeled within the occlusion period, and assign new tracks if the re-correlation score $s_{recorrelate}$ is higher than $s_{ReCthreshold}$.

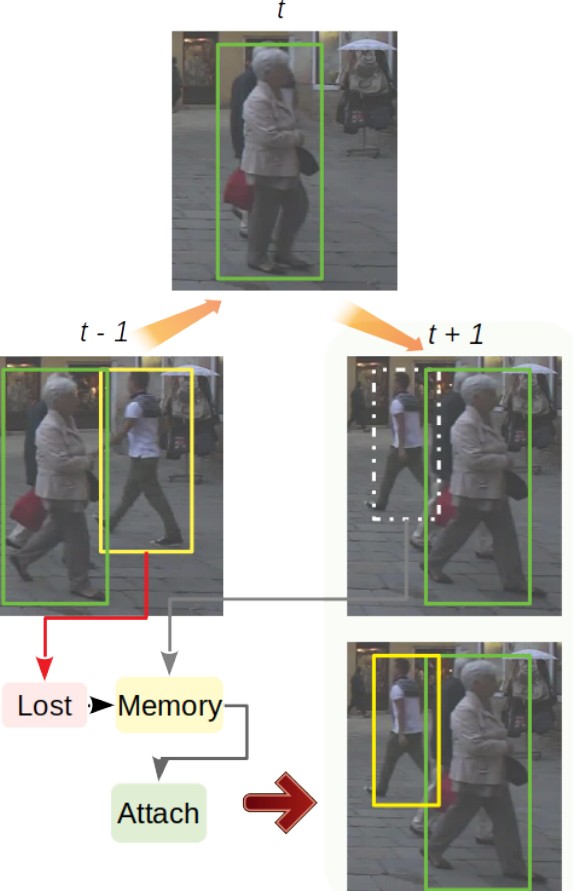

**Figure 4.** An example of how attaching handles object retrieval after occlusion. The occlusion happens at time *t*, the single object tracker belonging to the object in the yellow box at time *t* failed to track and save to memory. A new object in the white dotted box is initialized from detection at time *t* + 1 after occlusion. The attaching phase is able to attach the new object in the white dotted box to the lost one in the yellow box, eliminate the object in white dotted box, and keep tracking using the single object tracker from the yellow box object from before.

### 3.3.3. Re-Claiming

The third part, re-claiming, is a core procedure to retrieve occluded objects after messy frames as in Figure 5. For all the objects in the shadow tracking mode, as lost verification of detection for recent frames, the re-claiming part also performs a second-time correlation between newly initialized tracks and shadow objects. Moreover, the model calculates the overlap rate between those new tracks and the shadow object. If the re-correlation score $s_{recorrelate}$ and the overlap rate show that the new track is the doppelganger of the shadow object with a high probability, then the re-claiming phase chooses a more accurate bounding box from both tracks based on their tracking scores before, following a rule that each object can only appear a single time at a certain frame. One thing worth mentioning is that these kind of doppelgangers may be created by deformation or partial occlusion. This procedure can be viewed as a precaution in case of failures by the re-initiation part.

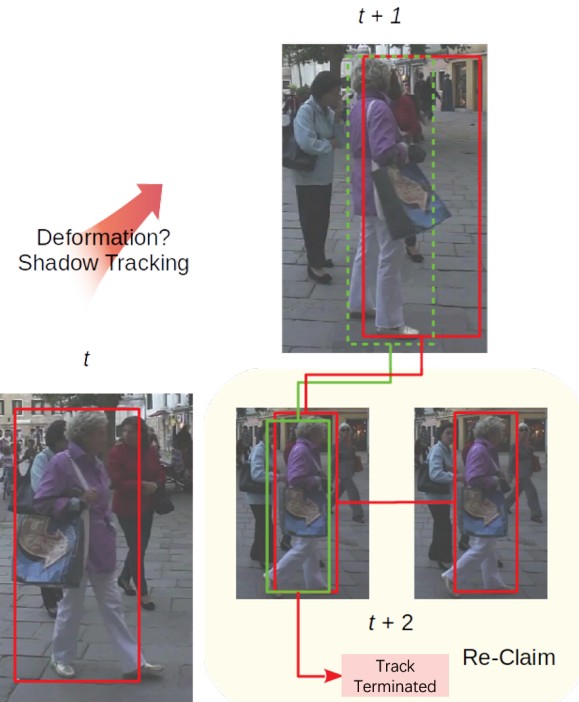

**Figure 5.** Demonstration of the re-claiming part. When encountering deformation, the single object tracker occasionally keeps on tracking using the trained model with non-deformed appearance features, which then will produce the inaccurate red box at time $t + 1$. At the same time, the corresponding detection which failed to align with a certain object might lead to the initialization of a new object in the green box. Under this circumstance, the re-claiming part chooses the more accurate bounding box and terminates the newly initiated object.

### 3.4. Initialization and Termination

The single object tracker heavily relies on the quality of the initialization bounding box in the first frame, which is usually hand picked in a SOT scenario. However, object candidates provided by an external detector under the current MOT framework are the only inputs to count on, though a high percentage of detections are imperfect in location and scale. We follow the argument in [6], that trackability of an object is not only dependent on its visibility, but also its size. After analyzing visibility and size of each external detector, we set the confidence score threshold for initialization as $\delta = -1, 0.6, 0.7$ corresponding to Deformable part model (DPM), Faster Region-Convolutional Nueral Network (FRCNN) and Scale-Dependent pooling (SDP). The single object tracker is initialized using the detection results in set $D_t$ left after pre-processing by the detection purification and filtering by $\delta$. As for track termination, since occluded objects will be saved in memory at every frame and trying to be re-established by the above procedures, the ones that can not be relinked for $t > 55$ frames are considered lost and removed from the memory to save computational resources.

### 3.5. Detection Purification

Detection plays an critical role in the tracking-by-detection paradigm. It is even more important when extending SOT methods to MOT as SOT relies much more on the quality of the detection results. Following the analysis of the detection in Section 3.4, we perform a purification strategy on the detection set $\mathbb{D}^{\approx}$. Firstly, we adopt non-maximum suppression (NMS) on the incoming frame. Then, we perform a strategy based on the assumption that detections in set $\mathbb{N}_j^t$ (which is the set of detections

within the neighborhood of the *j-th* detection) with a similar size, should be provided a similar score by the same detector as in the following:

$$b_{det} = \alpha \cdot b_{D_j}, where\ \alpha = \begin{cases} 1 & if\ size_j \in size_* \cdot (1 \pm \epsilon), (*) \in \mathbb{N}_j^t \\ 0 & if\ size_j \notin size_* \cdot (1 \pm \epsilon), (*) \in \mathbb{N}_j^t \end{cases}. \tag{6}$$

Here, $\epsilon$ is set to 0.2. With the two-step aforementioned cleaning strategies, detection results with low confidence and incomplete bounding boxes will be ruled out by the tracker.

## 4. Experiments

We perform an experimental study of the aforementioned models and further show that our models can tackle drift results when extending SOT to MOT, while maintaining tracking performance. We then demonstrate the tracking performance of our proposed method on the MOT Challenge dataset focusing on pedestrian tracking.

### 4.1. Implementation Details

We adopt the efficient convolution operator (ECO) tracker [3] as our single object tracker. ECO addresses computational complexity and over-fitting problems in state-of-the-art discriminative correlation filters (DCF) trackers by presenting a factorized convolution operator and a compact generative model of the training sample distribution. Due to the significantly increasing computational complexity when applying a single object tracker to a MOT scenario, we choose the faster version of ECO using hand craft features (histogram of oriented gradients (HOG) [31] and colour names (CN) [32]).

The proposed method is implemented in Python running on a desktop with Intel® Core™ i7-3930K @ 3.20GHz CPU. With the implementation of the binary-channel verification model and the delay processing model, the average speed of the proposed method on the MOT17 dataset is about 1.72 fps, which is acceptable compared with other state-of-the-art methods including LSST17 (offline) [33] at 1.5 fps, and DMAN (online) [34] at 0.3 fps. The average densities on each frame of the MOT17 training set and test set are 21.1 and 31.8 persons per frame, respectively.

The widely accepted clear MOT metrics [35] are adopted to evaluate the performance of our method, including multiple object tracking accuracy (*MOTA*) and multiple object tracking precision (*MOTP*), computed from false positives (*FP*), false negatives (*FN*) and the identity switch (*IDs*) as Equation (7). The calculation of aforementioned metrics are as follows:

$$MOTA = 1 - \frac{FN + FP + IDs}{GT} \in (-\infty, 1] \tag{7}$$

$$MOTP = \frac{\sum_{t,i} d_{t,i}}{\sum_t c_t}. \tag{8}$$

Here $d_{t,i}$ is the bounding box overlapping hypothesis $i$ with its assigned ground truth object, and $c_t$ denotes the number of matches in frame $t$. The *MOTA* score takes into account the number of times a tracker makes an incorrect decision. However, under some circumstances the ability to track the identities of objects is also worth measuring. In [35] it is introduced the mostly tracked (*MT*),

mostly lost (*ML*), and ID F1 (*IDF1*) scores to measure the performance of a tracker for those abilities. The computation of the *IDF1* score is formed as follows:

$$IDP = \frac{IDTP}{IDTP + IDFP} \tag{9}$$

$$IDR = \frac{IDTP}{IDTP + IDFN} \tag{10}$$

$$IDF1 = \frac{2}{\frac{1}{IDP} + \frac{1}{IDR}} = \frac{2IDTP}{2IDTP + IDFP + IDFN} \tag{11}$$

Here, *IDTP* is the number of true positive ID matches. This can be seen as the percentage of detections correctly assigned in the whole sequence. *IDFN* is the number of false negative ID matches, and *IDFP* denotes the sum of false positive IDs.

*4.2. Dataset*

We adopt the multi-object tracking benchmarks MOT17 to evaluate our tracking performance. It consists of several challenging pedestrian tracking sequences, with a significant number of occlusions and crowded scenes, varations in angle of view, sizes of objects, camera motion, and frame rates. MOT17 has the same video sequences as the lastest MOT16 [23] challenge benchmark, but provides more accurate ground truth in the evaluation. In addition to DPM [24], Faster-RCNN [25] and scale dependent pooling (SDP) [26] detections are also provided for evaluating the tracking performance. The number of trajectories in the training data is 546 and the number of total frames is 5316. The complexity of the tracking problem requires several metrics including multiple object tracking accuracy (*MOTA*), most tracked (*MT*), ID F1 score (*IDF1*) and so on. Specifically, the *MOTA* and *IDF1* scores quantify two of the main aspects: object coverage and identity. Also, we perform all the experiments using the public detections provided by the MOT Challenge for fair comparison. The single object tracker of our method is only initialized and online trained from public detection bounding boxes.

*4.3. Tracking Performance*

4.3.1. Evaluation Results on the MOT17 Datasets

Evaluation of our method is performed on the test set of the respective benchmark, without any training or optimization on the training set. The overall results accumulated over all sequences are shown in Table 1, including three sets with different public detectors in MOT17. Although our method does not show the state-of-the-art *MOTA*, high performance in both *IDF1* and *MT* metrics proves that our method can manage object ID consistently. This is what we aimed for. Specifically, most tracked (MT) and identity preserving (*IDF1*) (which compares groundtruth trajectory and computes trajectory by a bipartite graph, and reflects how long of a object has been correctly tracked) prove the effectiveness of our method on ID preserving, which outperform the other online trackers on MOT17. Different from online methods, offline methods do have both future and past information to further optimize the status of each object, which is usually expressed by a better overall performance. Despite this, our method still shows competitive ability on ID preserving, which is reflected by the *IDF1* and *MT* results. The overall results prove the ability of our methods to preserve the identity of objects. By adding another location estimation, provided by the single object tracker, as a second verification, the quality of the objects' representation is increased. Also the delay processing model helps in tackling the occlusion problem, which leads to better maintaining of the objects' identity after occlusion or deformation.

**Table 1.** Tracking performance on the MOT17 benchmark dataset. Best in bold. Definitions of abbreviations haven't been introduced before: ML, FP, FN, ID Sw. and Frag. stand for Most Lost, False Positive, False Negative, Identity Switch and Fragments respectively

| | | | | | | | | | |
|---|---|---|---|---|---|---|---|---|---|
| **MOT17 Dataset** | | | | | | | | | |
| **Mode** | **Method** | **MOTA ↑** | **IDF1↑** | **MT↑** | **ML↓** | **FP↓** | **FN↓** | **ID Sw.↓** | **Frag↓** |
| Offline | jcc [36] | 51.2 | 54.5 | 17.1% | **35.4%** | **20148** | 252531 | 2285 | 5798 |
| | eHAF17 [37] | 51.8 | 54.7 | **23.4%** | 37.9% | 33212 | 236772 | 1834 | 2739 |
| | SAS_MOT17 [38] | 44.2 | 57.2 | 16.1% | 44.3% | 29473 | 283611 | 1529 | **2644** |
| | eTC17 [39] | 51.9 | 58.1 | 23.1% | 35.5% | 36164 | 232783 | 2288 | 3071 |
| | LSST17 [33] | **54.7** | **62.3** | 20.4% | 40.1% | 26091 | **228434** | **1243** | 3726 |
| Online | HAM_SADF17 [2] | 48.3 | 51.1 | 17.1% | 41.7% | 20967 | 269038 | **1871** | **3020** |
| | AM_ADM17 [40] | 48.1 | 52.1 | 13.4% | 39.7% | 25061 | 265495 | 2214 | 5027 |
| | Tracktor17 [6] | **53.5** | 52.3 | 19.5% | 36.6% | **12201** | **248047** | 2072 | 4611 |
| | MOTDT17 [6] | 50.9 | 52.7 | 17.5% | **35.7%** | 24069 | 250768 | 2474 | 5317 |
| | DMAN [34] | 48.2 | 55.7 | 19.3% | 38.3% | 26218 | 263608 | 2194 | 6378 |
| | Ours | 48.9 | **57.0** | **21.9%** | 38.5% | 32914 | 253059 | 2392 | 4973 |

To further illustrate the effectiveness of our method on results refinement and object ID preserving, we carefully select other online methods with a similar number of *FP* and *FN* (which leads to a similar *MOTA* score) with ours, since ID consistency metrics are critically affected by FP and FN. According to Table 2, our method shows the best *IDF1* and *MT* results with the same level of *FP* and *FN*, which proves our method's ability of maintaining the objects' identities in a similar environment. As for ID switches metrics, our method only achieve the second best results in Table 2, which is because most of the switches occur when using the SDP detector, which has additional small detections. For smaller detections, feature representation of the single object tracker is coarse. Thus, correlation filters do not perform well when handling occlusion and deformation. Since our method performs online tracking, using future information to refine trajectories is not an option either.

**Table 2.** Comparison of ID consistency with similar MOTA (FP and FN) in MOT17 benchmark dataset.

| | | | | |
|---|---|---|---|---|
| **Comparison of ID Consistency** | | | | |
| **Method** | **MOTA↑** | **IDF1↑** | **MT↓** | **ID Sw.↓** |
| PHD_GM [42] | 48.8 | 43.2 | 19.2% | 4407 |
| MTDF17 [41] | 49.6 | 45.2 | 18.9% | 5567 |
| GMPHDOGM17 [43] | **49.9** | 47.1 | 19.7% | 3125 |
| HAM_SADF17 [2] | 48.3 | 51.1 | 17.1% | **1871** |
| Ours | 48.9 | **57.0** | **21.9%** | 2392 |

Figure 6 shows some of the visualization tracking results of our method under different circumstances. Specifically MOT17-03 and MOT17-08 represents static camera and crowded surveillance, while MOT17-07 and MOT17-14 show a moving camera. Each frame is chosen for their complexity that can show the performance of our method well.

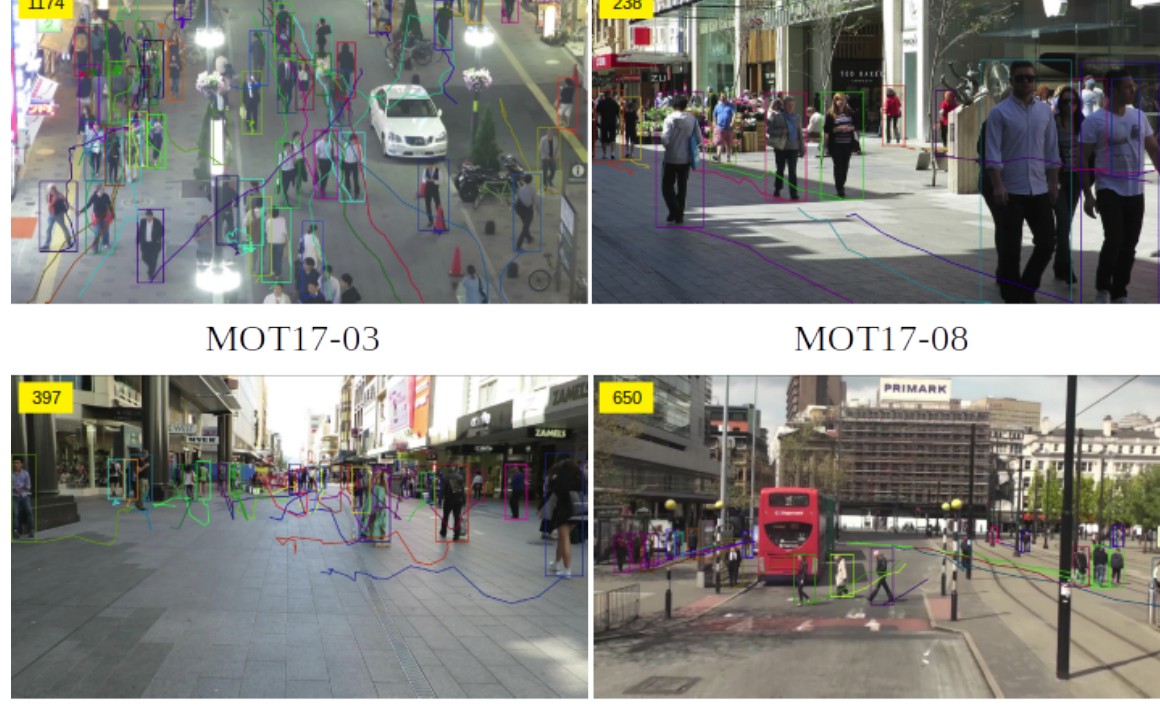

**Figure 6.** Visualization of selected sequences from the MOT17 benchmark dataset.

*4.4. Ablation Study*

For better understanding the effectiveness of each building block of our method, we ablate each part from the whole method and compare with a baseline which directly introduces the single object tracker within the MOT scenario , marked as the baseline in each table below. The analysis of performance is carried out on the MOT17 benchmark train set, and each study is conducted separately on the different external detectors (DPM, FRCNN, and SDP) provided by the dataset.

4.4.1. Binary-Channel Verification for Bounding Box Refinement

We show the effectiveness of our binary-channel verification (BCV) model within our proposed method based on the aforementioned baseline in Figure 7, denoted by BCV. We select the *MOTA* and *IDF1* metrics to evaluate the overall performance and the ID preserving ability. For both metrics, BCV shows significant improvements of performances on the DPM detector, while there is a relatively small cap between using the detection only and our BCV model on the FRCNN and SDP detectors. The reason for this is most likely the different quality of detection results provided by each detector. The DPM detector tends to have much more noise and usually produces partial detection under complex scenes, such as crowded and occluded environments for objects. On the other hand, BCV is able to increase both metrics for our baseline method, which is the single object tracker using only, where detection is only used to retrieve the object when occlusion happens. By introducing BCV, a clear increase of *IDF1* and *MOTA* shows that detection can help the tracker not only to better recognize each object during tracking, but also to optimize the training samples for the single object tracker that may lead to maintaining the identity of each object as long as possible. In this way, detection and the single object tracker may easily complement each other to achieve a better performance than using only one of them.

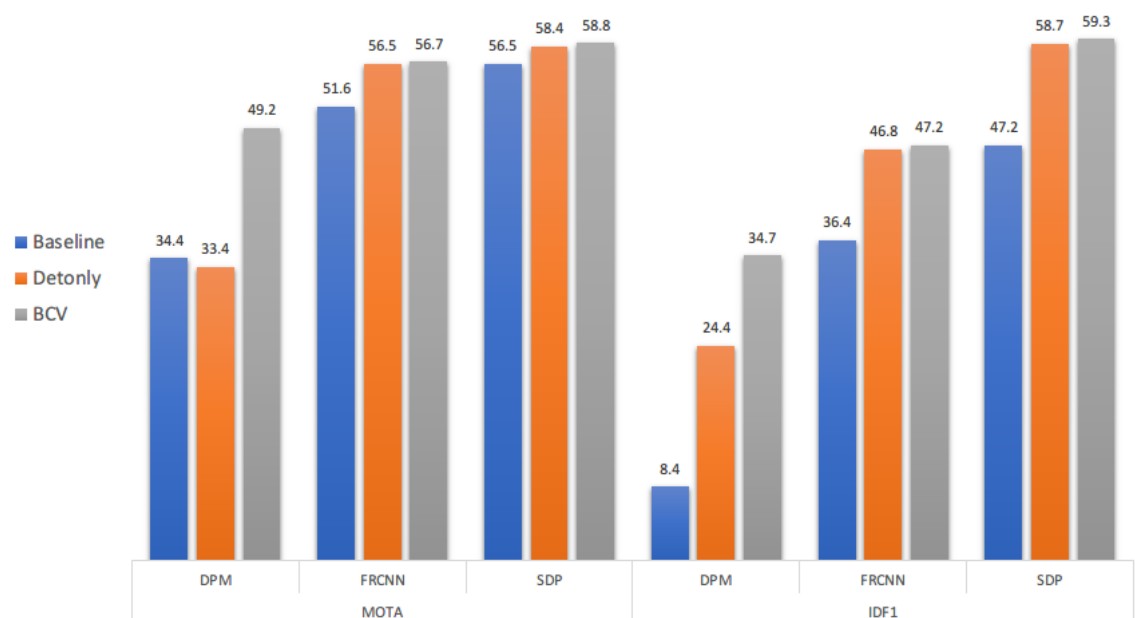

**Figure 7.** *MOTA* and *IDF1* scores of our binary-channel verification model. We separate detection from a different external detector such as DPM, FRCNN, and SDP. Blue columns represent our baseline which is directly using the single object tracker under the MOT scenario, while orange ones are the results using detection only and discard all estimation from the single object tracker except for the ones occluded or undetected and grey ones are our binary-channel verification model results.

### 4.4.2. Delay Processing Model

We separate the three parts of the delay processing (DP) model, and using the same metrics mentioned before to analyze how much each part contributes to the overall performance. Quantitative results of each part only along with integrated model are shown in Figure 8, where baseline indicates simply applying SOT in MOT, the same as in Figure 7. From Figure 8a–c, we can see that different parts of the DP model shows approximately the same effect on the baseline method, the quantitative increment of each part is more of the same. Base on that, one can easily presume that merging those three parts together should improve the performance enormously. However, as shown in Figure 8d, the overall performance of our DP model just slightly increases the quantitative results. That is because although these three parts are for three different circumstances, these circumstances can happen to the same object and emerge as different situation for the following frames. For instance, facing occlusion, one may continue tracking in shadow mode and cause a doppelganger problem, or completely stop, or even deform. Thus, either one part of the DP model could have solved the situation after several frames when we separate three parts, still tracking the same object as a result. Though the overall performance does not show a promising result as we expected, each part is non-trivial for the whole DP model, like icing on the cake. One can fill in the blank space while others ignore.

### 4.4.3. Detection Purification

Detection quality yields an important factor which influences the performance of the single object tracker , since it is critical during initiation and online training throughout the whole tracking procedure. By separating our detection purification strategy with the proposed method, we can tell the effectiveness of the detection quality in tracking. Figure 9 shows the results of our purification strategy for detection. Following the argument in [6], that the SDP detector has additional small detections and thus has less *FN* samples, which boosts the single object tracker since the tracker

itself can impossible initialize from a location that the external detector is incapable of providing. We can see this purification strategy affects most the DPM detector which shows the necessity of our detection purification.

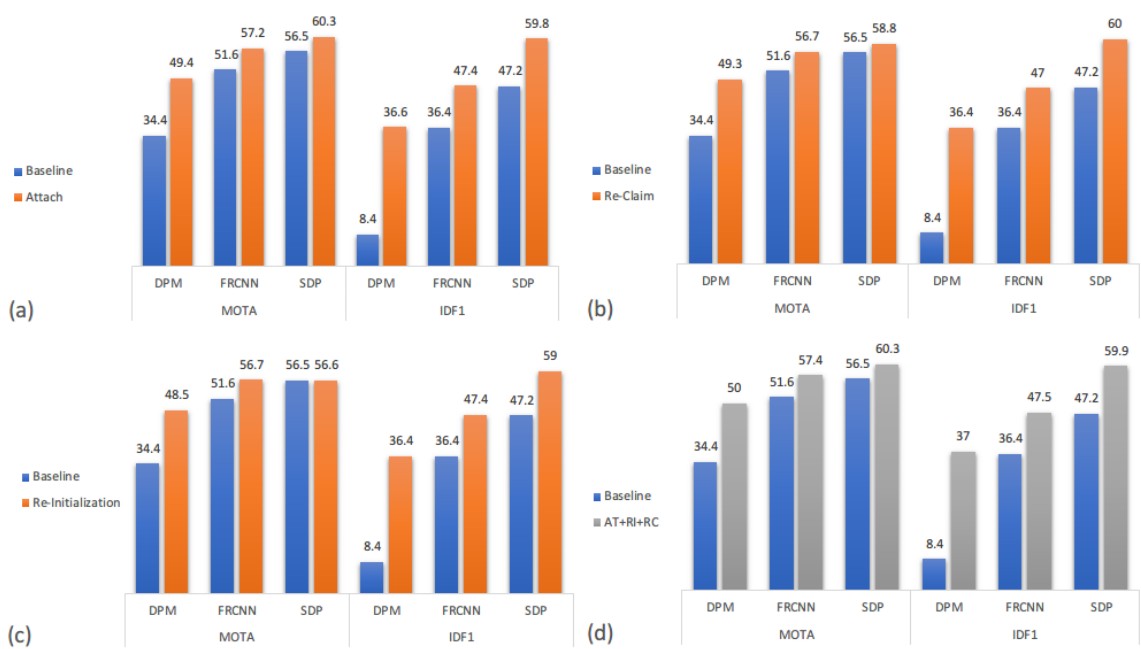

**Figure 8.** (**a**–**d**) show the comparison between each block and baseline method in *MOTA* and *IDF1* score respectively.

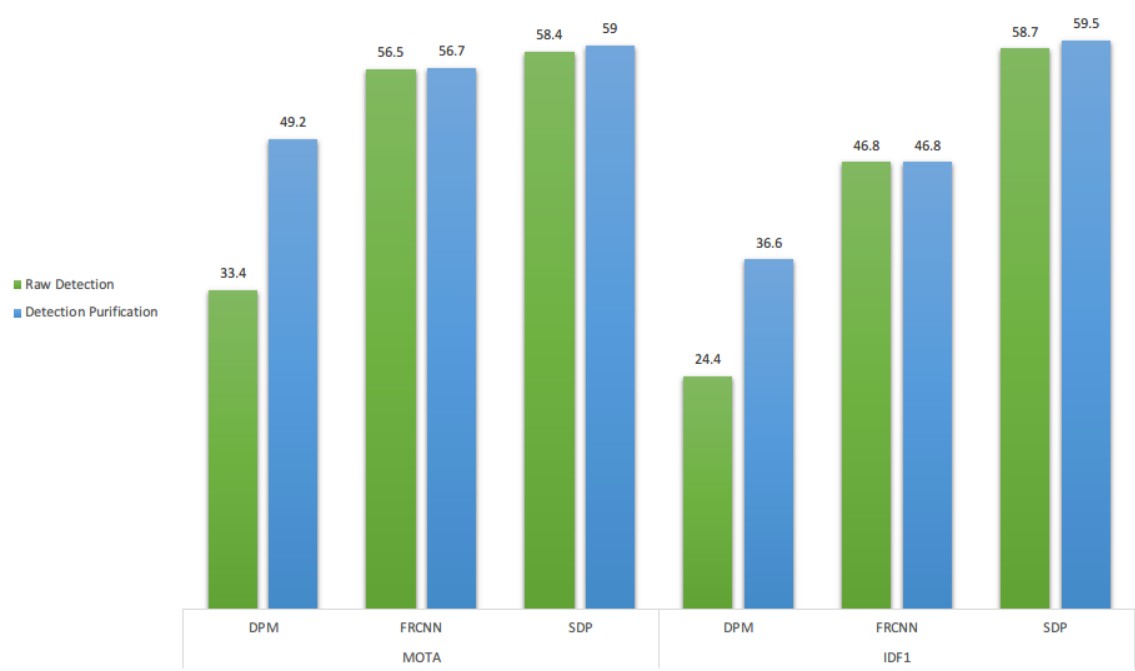

**Figure 9.** *MOTA* and *IDF1* change after applying detection purification.

## 5. Conclusions and Future Work

In this paper, we proposed a SOT-based MOT method in order to increase the accuracy of tracking results focusing on ID preservation. We introduce a binary-channel verification model using a separate single object tracker and detection provided by an external detector equally. Moreover, a delay processing model is proposed to handle the drift problem that easily emerges when applying a single object tracker under a MOT scenario. Our method outperforms other state-of-the-art methods on the MOT17 benchmark according to the *IDF1* and *MT* metrics, which proves our method is capable of preserving identity consistency when dealing with occlusion. Finally, we have shown some qualitative results under different hard-to-solve circumstances.

Though a re-initialization part is implemented in the delay processing model to handle geometry changes of certain objects, tracking objects with significant deformation after occlusion still remains unsolved since appearance features are mostly dissimilar for a tracker to reclaim the object. In our future work, we plan to train a generative model for the tracker to better recognize deformations of the same object without misalignment with other distraction information including similar objects.

**Author Contributions:** Conceptualization, M.L.; Data curation, M.L.; Formal analysis, M.L.; Funding acquisition, Z.M.; Methodology, M.L.; Resources, A.K.; Software, M.L.; Supervision, X.H., Z.W. and A.K.; Validation, M.L.; Writing—original draft, M.L.; Writing—review and editing, X.H., Z.W., J.W., Z.M. and A.K.

**Funding:** This research was funded by Jilin Scientific and Technological Development Program, grant number: 20180201013GX.

**Conflicts of Interest:** The authors declare no conflict of interest. The funders had no role in the design of the study; in the collection, analyses, or interpretation of data; in the writing of the manuscript, or in the decision to publish the results.

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
