# Peer review of "Enhanced Multiple-Object Tracking Using Delay Processing and Binary-Channel Verification"

_applsci, doi:10.3390/app9224771_

Round 1

Reviewer 1 Report

The authors present an interesting extension of current object tracking methods to multiple object tracking with promising results. The paper should be of interest to readers, but a few changes should be made to the paper before publication.

First, there are several issues with the use of acronyms. Please do not use acronyms before they are defined. Therefore, the title should not have these acronyms alone without their definition. Also, there are several misuses of SOT and MOT throughout the paper. For example, "tracker" is in the acronym, so saying SOT tracker is like saying single object tracker tracker. Please correct these issues.

Second, there are several language issues with the paper. Please review carefully before submission. A few examples are below.

On line 14, there should be a colon separating "three parts" and the items listed.

On line 40, "intuitively to implement" is not correct.

At the end of line 100, "to the" is not correct.

On line 250, "worth to mention" should be "worth mentioning".

Also, please format the references so that they are consistent.

Finally, there are some issues with the figures. In figure 1, there is a bounding box around two humans and the next stage text says "terminate". Please specify that you mean you will terminate the track of the human. Similarly in figure 4, you have "kill" but not what you plan to kill, which I'm assuming is the track.

Author Response

Dear Editor and Reviewers,

Best,

Li, Muyu

Reviewer 2 Report

The paper is well written and illustrated. The technical work appears of quality. Some short remarks:

Introduction could define in a better way the principles behind concepts MOT, SOT, etc.. This can enlarge the readability for persons outside the field  Explanations could go more in depth (because this and this ..) Few sentences can be improved  (line 37, 216, 299, 409-418)

Author Response

(The authors gave the same response as above.)
